# Data Fusion Analysis and Synthesis Framework for Improving Disaster Situation Awareness

Mehmet Aksit [1,*], Hanne Say [2] 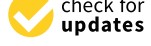, Mehmet Arda Eren [2] and Valter Vieira de Camargo [3]

1   Department of Computer Science, University of Twente, 7522 NB Enschede, The Netherlands
2   Graduate School of Engineering and Science, Ozyegin University, Istanbul 34794, Turkey;
    hanne.say@ozu.edu.tr (H.S.); arda.eren@ozu.edu.tr (M.A.E.)
3   Computing Department, Federal University of São Carlos (UFSCar), São Carlos 13565-905, Brazil;
    valtervcamargo@ufscar.br
*   Correspondence: m.aksit@utwente.nl

**Abstract:** To carry out required aid operations efficiently and effectively after an occurrence of a disaster such as an earthquake, emergency control centers must determine the effect of disasters precisely and and in a timely manner. Different kinds of data-gathering techniques can be used to collect data from disaster areas, such as sensors, cameras, and unmanned aerial vehicles (UAVs). Furthermore, data-fusion techniques can be adopted to combine the data gathered from different sources to enhance the situation awareness. Recent research and development activities on advanced air mobility (AAM) and related unmanned aerial systems (UASs) provide new opportunities. Unfortunately, designing these systems for disaster situation analysis is a challenging task due to the topological complexity of urban areas, and multiplicity and variability of the available data sources. Although there are a considerable number of research publications on data fusion, almost none of them deal with estimating the optimal set of heterogeneous data sources that provide the best effectiveness and efficiency value in determining the effect of disasters. Moreover, existing publications are generally problem- and system-specific. This article proposes a model-based novel analysis and synthesis framework to determine the optimal data fusion set among possibly many alternatives, before expensive implementation and installation activities are carried out.

**Keywords:** disaster situation awareness; UAVs and data sources; quality of data fusion; model-based framework for determining optimal data fusion; domain model of data sources for earthquake detection; automated synthesis for data fusion

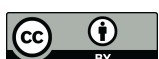

## 1. Introduction

We define disaster situation awareness as the ability of the authorities to effectively and efficiently detect the negative effects of disasters so that aid operations can be planned and executed in a timely manner. In general, the objective is twofold: to understand the type and magnitude of the damage caused and to determine the conditions and locations of the persons that need help.

The concept of *situation awareness* has been studied extensively and applied in several areas [1] such as disaster management [2]. Unmanned aerial vehicles (UAVs) (also known as drones), for example, can be used to detect the effect of disasters [3]. UAVs flying over a disaster area can take, compare and analyze images obtained before and after the disaster. Although UAVs can be considered adequate for some purposes, they may fall short of detecting certain facts such as the location of persons under rubble. Moreover, UAVs may take a considerable amount of time before their missions are completed. Nevertheless, during the last decade, we have observed the introduction of the concept of so-called *advanced air mobility* (AAM) and its implementation by unmanned aerial systems (UASs), where swarms of UAVs cooperate together for a common mission [4]. All these new technologies help in creating more effective disaster management systems.

Additionally to the use of UAVs, disaster-related data can be gathered from various sources. For example, dedicated sensors can be attached to physical objects such as residences, bridges, and roads to detect if the corresponding structure is damaged [5]. Base stations of mobile network providers may supply information about the location and use of mobile phones [6]. Dedicated systems can be brought to the disaster area, such as microwave radars to determine if there are living persons under rubble [7]. Official registration databases can be consulted to estimate the possible victims of the collapsed residences. For brevity, in this article, such devices and systems are abstracted as data sources.

Emergency control centers aim at minimizing the negative effects of disasters by detecting and monitoring disaster situations and by carrying out aid operations accordingly [8,9]. The success of these centers depends on the accuracy and timeliness of the gathered data. There is a large set of possible data sources, each with its advantages and shortcomings.

Data fusion [10] is a promising technique to get the best out of multiple data sources. Due to high investment costs, before designing and implementing data fusion systems, it may be beneficial to estimate the optimal set of data sources that give the best combined effectiveness, cost, and timing values of sensor fusion. This requires a set of tools for the analysis and/or synthesis of heterogeneous data fusion systems before they are installed. Implementation alternatives of data sources and fusion techniques are considered out of the scope of this article.

Unfortunately, there are almost no publications devoted to analysis and synthesis of prospective data fusion configurations in disaster/earthquake management. Moreover, most proposed solutions are problem- and/or system-specific. What is needed is a framework which enables us to define models for a large category of data fusion alternatives. These models can for example be formed manually by an expert or be computed by synthesis algorithms. The framework should be extensible to introduce new models of geographical elements, data sources, and analysis and synthesis algorithms.

The contributions/novelties of this article are as follows. First, to detect disaster situations, a novel domain model is defined for representing the relevant data sources. In this article, earthquakes are chosen as an example of disaster. Second, to represent geographical areas, an object-oriented model is defined. In addition, dedicated queries are introduced to create models of data fusion associated with a selected set of geographical entities. Third, a model-based framework is introduced to specify the candidate data sources for a given geographical area. Fourth, with the help of the framework, the effect of various alternatives of data fusion can be estimated. Last, to synthesize the optimal set of data sources within specified constraints, algorithms are defined. To the best of our knowledge, such a framework has not been proposed by the research community before.

This article is organized as follows. The following Section 2 introduces the background and related work. Section 3 presents the problem statement, the research questions, and the method adopted. Section 4 describes the model base for data sources, presents the domain model, and gives examples of a selected set of data sources. The architecture of the framework and the associated object-oriented models are explained in Section 5. Section 6 defines the objectives of data fusion and presents formulas to calculate a selected set of quality values of the user-defined data fusion models. Two synthesis algorithms are presented in Section 7. Section 8 describes how the proposed framework can be extended to support UAS-based data fusion systems. Section 9 discusses the threats of validity of the assumptions made in this article. Section 10 presents our research plans. Finally, Section 11 gives the results and conclusions. The Appendices A and B present the estimated characteristics of the data sources which are referred to in this article.

## 2. Background and Related Work

### 2.1. Situation Awareness

From a *systems* perspective, *situation awareness* can be defined as enabling systems to sense, adapt, and react, based on the environment. During the last decades, within various application domains, a considerable number of research publications has been presented on

situation awareness such as military [11–13], disaster management of different kinds [2,14], smart manufacturing [15], tourism [16], connected cars [17], and advanced aerial mobility applications [18]. In [1], situation awareness is studied conceptually from the perspective of human behavior.

The publications on situational awareness can be classified from several perspectives. Although there are similarities, the application domain is the one of the determining factors in the way a system with situation awareness is designed. Most publications on situation awareness are, therefore, 'problem specific' [11–18]. For example, in [13], conceptual models are presented to evaluate the quality of symbolic information, and to semantically integrate heterogeneous and dynamic environments with cyberspace exploration. The initiatives presented are interesting but are specific to defense and security domains. A considerable effort may be needed to adapt the proposed techniques to reduce disaster situation awareness problems.

Furthermore, the majority of publications propose a particular system and discuss how the system accomplishes its design objectives. These publications are 'system specific'.

Models for representing environments are considered important in the way data are gathered and interpreted. In this context, for example, data fusion techniques are evaluated [15]. Various analysis methods are proposed, for example, based on statistical analysis [19], Bayesian networks [20], and ontology-based techniques [21]. Multiplicity of data sources, such as crowd-sourcing, sensors, UAVs, and fusion of these are also investigated [22].

### 2.2. The Disaster Situation Awareness Problem

The disaster situation awareness problem is a special case of the general situational awareness problems discussed in the previous section. Immediate and accurate detection of disaster situations is crucial for an effective and efficient response. Unfortunately, depending on the kind, size, and intensity of disaster situations, emergency control centers may have difficulties in obtaining the necessary information as needed. We termed this problem the disaster situation awareness problem. There have been a considerable number of research publications on earthquake prediction, but unfortunately, on-time prediction is still a difficult problem [23]. Although this article focuses on the data sources necessary for after an earthquake period, naturally, the necessary data fusion system must be planned, designed, and installed before a disaster occurs.

When a disaster occurs, emergency control centers try to gather information from all kinds of sources such as victimized persons, existing sensors, telephone calls, authority reports, UAVs, and satellite images [24–26]. There are a considerable number of publications on disaster/earthquake management [2,8,9,14,26]. To increase accuracy and efficiency of disaster situational awareness, data obtained from more than one data source must be fused and interpreted, preferably in an automated way.

One of the main problems in enhancing disaster situation awareness is the high investment costs. Firstly, to improve the awareness, it may be necessary to install many kinds and numbers of data sources within a large geographical area. Secondly, designing such a large-scale data fusion system is a challenging task due to the topological complexity of urban areas, and the multiplicity and variability of the available data sources. Therefore, automated tools are needed.

### 2.3. UAVs and Data Fusion Techniques

UAVs can be defined as small aircraft that can be operated remotely by pilots or programmed to operate without the assistance of humans. UAVs were originally developed for military purposes, but they are now a major focus of research in several disciplines. The technical characteristics of UAVs play a significant role in their categorization, such as technology used, level of autonomy, size, weight, and energy resources. UAVs are often equipped with a variety of sensors, including radars, television cameras, global posi-

tioning systems (GPS), satellite communications, image intensifiers, and infrared imaging technology [27].

Nowadays, UAVs are used in many application domains, such as: precision agriculture [28], construction and infrastructure inspection [29,30], and rescue of people [31] and also in the same context of this article—disaster management. For example, a literature review on the application of drones in the disaster management context is presented in [26]. The authors of that work have analyzed papers from 2009 to 2020 and classified them into four categories: (1) mapping or disaster management; (2) searching and rescuing; (3) transportation; and (4) training.

UAVs can be also prepared for specific missions such as face recognition. In [32], for example, a deep neural network model is presented to improve the performance of face detection and recognition tasks when photos are taken from high altitudes. However, in the case of earthquakes, detecting living persons under the rubble is an important concern, where no face information is available. Our article does not focus on the algorithms which can be adopted in the implementation of data sources. Nevertheless, such techniques can be adopted to improve the quality of data sources suitable for certain disaster types.

There are attempts to measure the concentration of gases and aerosols in the atmosphere by installing the necessary equipment on board commercial airliners [33]. The samples are collected with an inlet probe, which is connected to the equipment on board. This allows accurate sampling of aerosol and chemical contents. The samples are analyzed in ground-based laboratories. These data can be used, for example, in cases where forest fires have to be monitored. Such techniques can be linked to other data sources such as satellite data and to UASs ascending in situ.

It is stated that although there is a significant increase in the number of publications on use of drones in disaster cases, a limited amount of research is observed to address post-disaster healthcare situations especially with regards to disaster victim identification.

Another topic related to this paper is remote sensing [34], which is used to describe information that is gathered from distant targets. This is usually done by satellites, aircraft, and UASs. An interesting example of remote sensing using imagery taken from commercial flights is presented by Mastelic et al. [35].

Data fusion techniques are introduced to enhance the accuracy and reliability of multiple but related data sources [10,36,37]. In [37], for example, a survey of simultaneous localization and mapping (SLAM) and data fusion techniques for object detection and environmental scene perception in unmanned aerial vehicles (UAVs) is presented. The analysis performed by the authors revealed that a combination of data fusion and SLAM can assist in autonomous UAV navigation without having a predefined map. In this approach, raw sensor data are directly provided as an input to the SLAM algorithms. The data fusion process may be made more end-to-end using machine learning techniques. These techniques are suitable in enhancing the effectiveness and efficiency of data detection by UAVs, for example, to improve situation awareness.

Another related work is presented in [38], in which a survey on UAV orthoimage generation technologies that focus on mainstream frameworks is presented. The authors give a detailed comparison of the important algorithms by referring to their performances and propose extending the frameworks with deep learning techniques. Our focus in this article is more on the fusion of heterogeneous data sources but not on the techniques adopted in the implementation of individual data sources.

In the literature, data fusion techniques are classified as signal fusion and information/data fusion [39]. Signal fusion is used to combine information from multiple sources that measure the same type of physical phenomena, but with different characteristics. Information fusion combines data/information from multiple sources to provide a more accurate understanding of a situation. This article adopts information fusion techniques.

Due to the progress in sensor technologies and related systems, in addition to UAVs, it may be beneficial to adopt multiple data sources to enhance awareness of disaster situations.

None of the publications, however, propose analysis and synthesis methods for the purpose of determining the optimal set of data sources, before expensive design and implementation activities are carried out. Moreover, model-based frameworks are in general not supported, meaning that system architectures are fixed and when needed, they can not be easily extended, for example, by introducing new geographical and data source models, and different analysis and synthesis algorithms.

### 2.4. Advanced Air Mobility and Enhancing Disaster Situation Awareness

Recently, there has been considerable interest in the concept of *advanced air mobility* (AAM), which aims to provide safe and reliable on-demand aerial transportation for customers, cargo, and packages [4]. The UASs (unmanned aerial systems) may consist of swarms of systems with multiple UAVs, ground-stations, various data sources, autonomous vehicles, smart mobile devices, etc. [4,18,40].

Research topics include a combination of challenges from many disciplines, such as design of UAVs, situation awareness from a broad perspective, autonomous navigation and coordination, and safety and reliability issues.

UASs can bring additional opportunities for improving situation awareness for disaster/earthquake management. Firstly, UAVs may not only function as data sources but also as the possible data fusion nodes. Secondly, in addition to ground-based data sources, a swarm of UAVs can dynamically cover a large area for better situation awareness. Thirdly, UAVs can share certain responsibilities among each other in lifting some heavy equipment such as radars and precision optical devices.

To the best of our knowledge, analysis/synthesis of UAS-based data fusion systems for disasters/earthquakes have not been studied sufficiently in the research media. We will elaborate on these opportunities later in this article as a future work.

## 3. Problem Statement and Approach

Based on the literature study, we observe that the current research proposals on enhancing disaster situation awareness have one or more of the following shortcomings:

- There is a lack of a model-based framework where, with the help of model management tools, a large category of data sources and geographical elements can be defined.
- A domain model of data sources for earthquake detection is missing.
- There is a lack of analysis tools to evaluate various prospective data source fusion alternatives for the purpose of achieving higher effectiveness.
- A toolset for automated synthesis is lacking, which can help in finding out the best data fusion alternative for a given set of constraints.

### 3.1. Research Questions

To address these shortcomings, the following four research questions are elaborated:

1. What is a suitable architectural style of the desired model-based framework? How can this architecture be extended to deal with the (future) UASs?
2. How to define a domain model of the data sources suitable for detecting the effects of earthquakes?
3. How to compute the combined effectiveness, cost, and timing values as a result of data fusion? The algorithms for computation must be changeable and re-definable to satisfy different needs.
4. What kind of algorithms can be defined for automated synthesis of the optimal data fusion?

### 3.2. Method

Our approach adopts techniques from various disciplines such as *software engineering, model-driven engineering, programming techniques, and algorithm design* [41].

- Based on software engineering principles, architectural styles [42] are adopted.

- The proposed framework is inspired from model-driven engineering techniques [43]. Within this context, a domain analysis work is carried out and the 'feature-model' notation is adopted for representing the domain of data sources.
- From programming techniques and algorithm design, design patterns [44] are adopted. Patterns provide flexibility to the proposed framework. In addition, object-oriented programming and querying techniques are adopted for relating the candidate data sources to the elements of a geographical area. As for algorithms, data fusion formulas and optimization algorithms are implemented for data fusion synthesis and for computing the effectiveness, cost, and timing values of the fused data sources.

## 4. Domain Model for Data Sources

In this section, first a model of the domain of data sources which can be used for detecting the effects of earthquakes is given. Second, a selected set of data sources is described in more detail.

### 4.1. A General View of the Domain

To represent the relevant data sources in a diagram, we adopt the feature diagram notation [45]. This notation was defined to visually express the features of the domain of interest, and it consists of symbols to represent mandatory features, optional features, group, alternative group, abstract features, and concrete features. The diagram is in fact a visual representation of propositional logic formulas [46].

In Figure 1, depending on the attachment characteristics of the data sources, the domain is divided into three branches. The first branch represents the data sources which can be attached to geographical areas. The second branch represents the data sources which can be attached to physical objects. The third branch represents the data sources which can be transported to the locations where disasters have been effective.

The first branch consists of three sub-branches: Airborne, Aerial trackers, and Registration database. The second branch has four sub-branches: Disaster detector, Smart-home detector, Optical detector, and Smart utility meter. The third branch has three sub-branches: Microphone, Carbon dioxide meter, and Microwave radar. The gray-colored rectangles correspond to the concrete features. The white-colored rectangles must be refined into the concrete ones.

The feature model can be used to create a concrete model by using the following strategy:

1. Starting from the root node, select the compulsory features, if any.
2. Decide if the optional features must be selected, if any.
3. By obeying the defined semantics, refine the features which are grouped together by the logical operators of the feature model (for example OR, Alternative OR), if any.
4. Continue with this process from the abstract features towards the concrete ones until no optional and/or abstract feature is left unresolved.

In this strategy, it is assumed that all leaf features are concrete and no cross-tree constraints are specified. In Figure 1, the compulsory features are 'Attached to geographic area' and 'Registration database', since registration of inhabitants is generally enforced by law.

The model given in Figure 1 can be extended by adding new data sources or refining the existing abstract features. Additionally, there may be many commercially available data sources in the market with varying characteristics. These can be added to the figure as the leaves of the tree structure. We will now elaborate on these data sources in more detail.

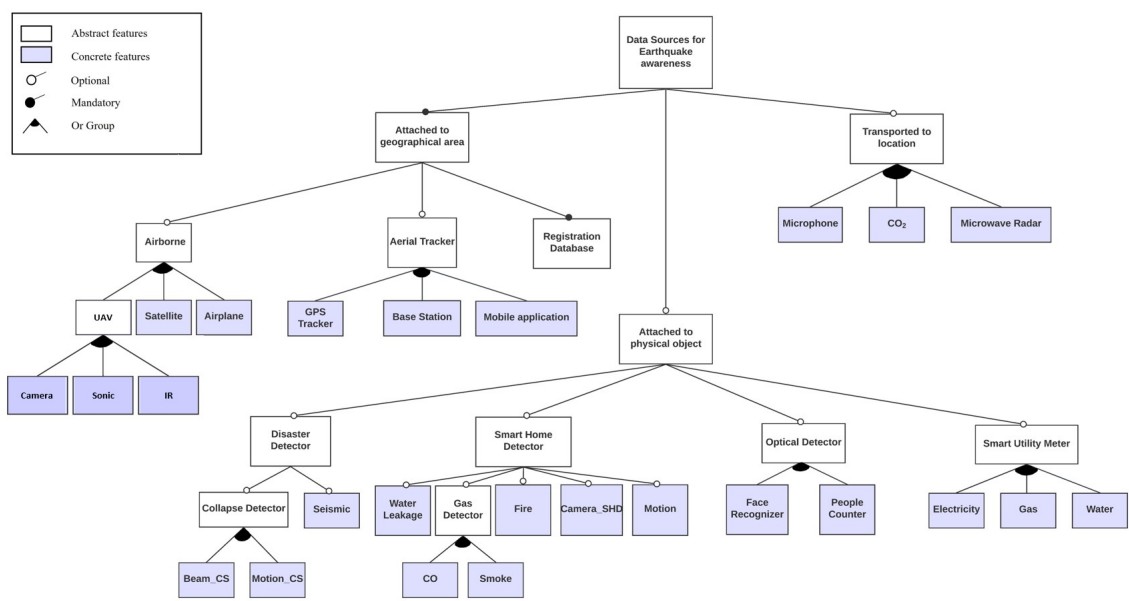

**Figure 1.** Feature model representing the domain of data sources for detecting the effect of earthquakes.

### 4.2. Data Sources Which Can Be Attached to Geographical Areas

**Airborne data sources:** UAVs, airplanes, and satellites [3,25] are considered in this context. They can be used to observe a large area by sending images or videos of the disaster area via a communication link to a ground station.

In this diagram, we consider three data sources for UAVs: Camera, Sonic, and IR. Optical cameras are especially relevant for estimating the effect of disasters such as earthquakes. They can have a high probability of detection especially during a daytime surveillance mission. Ultrasonic sensors (Sonic) can be used at low altitudes, for example, to measure the level of floods. Infrared sensors (IR) can be used to detect living creatures and moving targets such as vehicles from altitudes of 1 to 5 km.

Cameras and infrared sensors can also be attached to satellites and airplanes. It is assumed that the gathering time with airplanes is higher than that of UAV. The response time of a satellite depends on many factors, for example, whether it is geosynchronous or not, the location of the satellite, the availability of online tracking by the base station, etc. For brevity, the sub-features of satellites and airplanes are not shown in the figure.

**Aerial trackers:** Within this context, we consider GPS trackers [47], mobile phone applications [48], and base-stations [6]. These data sources are relevant for estimating the coordinates of persons. The accuracy of detection of positions of persons depends on where the tracked persons are located. A GPS tracker is useful in tracking position, direction, and time of movements. Mobile phone applications are also based on GPS; however, they use the internal sensors of a mobile phone and track position through an application. Base stations track positions based on online information about the registered mobile phones.

**Registration database:** Every well-organized municipality has a well-maintained list of inhabitants per location. This information can be used to estimate the existence of persons under the rubble after the occurrence of earthquakes. However, the probability of detection depends on many factors. In addition to a normal living pattern, a person may be living in multiple addresses, temporarily away from home, or even not living at that address at all. The probability may also depend on night- or day-time occurrences of the disaster.

### 4.3. Data Sources Which Can Be Attached to Physical Objects

**Collapse Detector:** The purpose of this data source is to determine whether a structure has been broken or not. A motion sensor detects the movement of an object at a given location. A beam sensor can measure whether the surface of a structure is straight or not.

**Seismic Sensor:** It may detect ground oscillation, and accordingly outputs a signal as a waveform [49]. This sensor can be useful in estimating the effect of earthquakes. It can accurately measure the intensity of an earthquake. However, to determine whether the structure has been damaged or not, one needs precise information about the robustness of the corresponding structure and the ground mechanics of the location.

**Smart home detectors:** These sensors are commonly used in households for the purpose of protecting the inhabitants from incidents such as water and gas leakage, smoke and fire, and burglary [50]. The usefulness of these data sources depends on the type of smart home detector. For example, consider the following sensors:

Carbon Monoxide sensor is designed to warn the user of any unusual build-up of CO in where it is located. Smoke sensor measures the presence of smoke and detects fire by sensing small particles in the air. Fire sensor detects fire according to the occurrence of flames by sensing light beams. Motion sensor is used for detecting motion around the sensor. Camera can detect motion, record, and give a warning.

Motion and camera detectors can be useful in estimating the presence of persons. One important issue with these sensors is that adopting them for disaster detection may interfere with the privacy of the inhabitants. We therefore assume that unless permission is given, these sensors can only be attached to open areas such as schools, shopping centers, hospitals.

**Optical detectors:** In this context, we consider face recognition systems [51] and people counters [52]. Face recognizer is a way of confirming an individual's identity using an online face image. To this aim, images of the persons to be identified must be pre-recorded by the system. With the help of pattern recognition algorithms, the online image is classified according to the recorded images. These systems may hep in estimating the locations of persons. In addition, the camera that is used for face recognition may also function as a disaster detector, if programmed accordingly. People counters are used to count the number of persons entering or leaving a location. This sensor can help in estimating the presence of persons.

**Utility meters:** In this context, smart electricity, water, and gas meters are considered [53]. These meters record information about the consumption of electrical energy, water, and/or gas. The gathered data can be used to infer the living patterns within a location and as such, they can help in estimating if anyone is present in a location just before the occurrence of an earthquake.

### 4.4. Data Sources Which Can Be Transported to Certain Locations

**Microphone, carbon dioxide meter, and microwave radar:** These data sources are generally used after earthquakes by transporting them to the location of the disaster. Microphones are brought under the rubble to detect any meaningful sounds. Carbon dioxide meters measure the breathing of living creatures [54]. Similarly, microwave radars [7] detect movements of persons and breathing of lungs. All these data sources are relevant for estimating the location of persons. Microwave radars are quite accurate and effective; however, they are much more expensive than the other two sensors.

**Other data sources:** For brevity, this article mainly focuses on data sources that can be used for earthquake detection. There are of course many more data sources which can be used for various purposes. For example, barometers and compasses are mainly used by the flight control systems of UAVs; such data sources are considered out of the scope of this article. The adoption of a mobile analysis laboratory in situ is a promising approach which is likely to be more often used in detecting the effects of disasters including earthquakes [33].

## 5. Architecture of the Framework

This section describes the framework that is designed for the analysis and synthesis of the prospective data fusion alternatives. The intention of this section is not to explain the implementation of the system in detail, but rather provide an architectural view and abstract technical description so that the reader may gain an insight into its structure and the working of it.

A symbolic representation of the architecture is given in Figure 2, using the 'UML component diagram' notation. Three stakeholders are shown in the figure: *Analyst*, *Data source modeler*, and *Geographical area digital-twin modeler* in 'actor' notation. The stakeholder *Designer of the system*, who is in charge of designing and implementing the framework, is not shown here. With the help of the component *UI*, the *Analyst* analyzes or synthesize the prospective sensor fusion alternatives. To this aim, *Data source modeler* defines the models of the available data sources using the feature-modeling tool, as explained in the previous section. *Geographical area digital-twin modeler* is responsible for creating a model of the geographical area that is considered. The *UI* component retrieves data from the components *Feature-modeling tool* and *OO database*. The component *OO database*, in turn, accesses the component *Digital-twin model of Geographical-area*. An external *GIS database* provides the necessary data for this purpose. The component *UI* utilizes the functions offered by the component *Model evaluator* for the analysis and synthesis operations. The component *Algorithms* offers the necessary algorithms for both analysis and synthesis. For dynamic menu generation, the components *Simulator* and *OO database* use the services of the component *UI*.

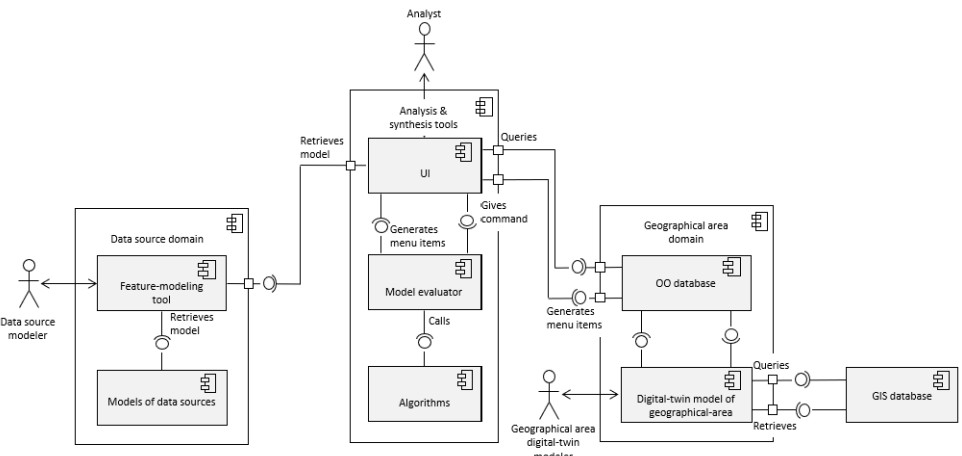

**Figure 2.** A symbolic representation of the architecture of the model-based framework.

The proposed architecture is 'model-based'. Without modifying the system, data source and geographic information system models can be defined and introduced into the system with the help of model management tools. In addition, UI menus, analysis and synthesis algorithms, and most system functionalities are designed as plug-ins; they can be replaced after the installation of the system without re-compilations.

The model of the geographic area is based on CityGML [55], which is an extensible Geographic Information System (GIS) specification. The models and meta-models of CityGML can be extended by subclassing the existing classes. However, it is claimed that the current models of CityGML are not expressive enough to represent the data structures required by disaster management systems, and to this aim, an extension of CityGML is proposed [56]. In this article, we adopt the proposed extension, where a rich set of element types are supported, such as Bridge, Factory, Financial Center, Firefighter Station, Historical Buildings, Hospital, Logistic Center, Residence, Restaurant, Road, School, and Shopping Center. In addition, disaster-specific data types are defined such as Emergency Control Center, Collapse, Fire, Flood, Landslide, and Tsunami.

Figure 3 shows a snippet of class Residence. The attributes of class Residence are grouped in three categories marked with the stereotypes constant, dynamic, and derived.

The attributes tagged with the stereotype «constant» are those whose values are set during the instance creation phase, when a digital-twin model [57] of an urban area is defined. Regarding the attribute *robustnessFactor*, it is defined by experts depending on the earthquake resilience properties of the construction and the ground mechanics of the construction site [58] . The value of this attribute can be set as 'undefined', or be defined within the range of 0 to 5.

The dynamic attributes are those whose values may change in time. For example, the attribute *nOfPersons* can be estimated, set, or modified by a simulator, or it can be based on an estimated number, derived from the relevant data sources.

The dynamic attributes whose names begin with the word *attached*, are responsible for storing the data sources that are attached to the corresponding instance of class Residence. As we are dealing with a modeling environment, the values of these attributes are objects that represent the data sources. They are also used for calculating the effectiveness, timing, and cost factors of the corresponding instance.

The dynamic attribute *queries* contain a set of 'command-query objects' which can be executed on the instances of this class. Each element class, such as class Residence, may have a dedicated set of queries, which can be stored and retrieved by calling on the corresponding 'setter and getter' methods. This provides run-time extensibility in the kinds of queries a class may provide.

**Figure 3.** An excerpt of the attributes and methods of class Residence.

*The Analysis Process*

The analyst may utilize the system, for example, in the following way:

1. Let us assume that a number of instances of class Residence has been created which represents a selected set of actual residences of an urban area under consideration. In this process, the constant attributes of these instances have been initialized as well.
2. Additionally, for the instances of class Residence, a set of dedicated 'command-query objects' has been defined and stored in the attribute *queries*.
3. At this stage, these instances are now ready for use to analyse a model of a prospective sensor fusion system. First, a fusion model must be defined.
4. With the help of the user interface (UI), the analyst observes these queries, which are displayed at the UI as menu items. An implementation of Command pattern [44] provides a dynamic menu generation mechanism.

5. To simulate the attachment of the prospective data sources, an appropriate query is selected by activating the corresponding menu item. The data source objects are also selected by using the feature-modeling tool as defined in Section 4.

6. With the help of an object-oriented database, and depending on its definition, the selected query item is executed over one or more instances of class Residence.

7. While executing the query, the database calls on the necessary 'setter methods' of the corresponding instances. To this aim, class Residence provides the necessary method interface.

8. Depending on the query, the selected data source objects are stored in one or more instances of class Residence.

In contrast to the constant attributes, the derived attributes are computed. Consider the following process as a follow-up step of the previous example:

9. When the data source objects are stored in an instance of class Residence, the method *calculateEffectiveness()* is called on 'self'.

10. This method retrieves the properties of the stored data source objects, and by using the formulas presented in this article, it computes the values of the derived attributes.

11. If a new set of data source objects is stored, the derived attributes are computed in the same way again.

12. Now assume that the analyst executes another menu item for attaching a UAV as a data source on the corresponding geographical area.

13. In our system, a geographical area is represented as an instance of class GeographicArea. The method *notifyAttach()* is automatically called on the corresponding instances of class Residence, when a data source, such as a UAV is attached to the corresponding geographical area. An implementation of Observer pattern [44] provides an 'event propagation' mechanism.

14. When called, this method reads the characteristics of this new data source, registers its identity, and calculates the derived attributes again. Similarly, the method *notifyDetach()* is used when the corresponding data source is removed from the geographical area.

The other implementation-related attributes and methods of class Residence are not shown for brevity. Of course, the process given in this section has the intention to clarify a possible usage of the instances of class Residence. In the implementation phase, depending on the language adopted, different coding alternatives may be used.

Class Residence is not the only modelled element in the system. All physical objects share certain properties such as constant, dynamic and derived attributes, a number of 'setter and getter' methods, methods to calculate the derived attributes, and methods which are used for notification purposes. Adopting a standard interface for all physical objects enables interfacing with the object-oriented database smoothly. The query objects per class may differ considerably from each other. Nevertheless, these objects have a uniform interface with the UI and the database. For brevity, we do not show the other class diagrams in this article.

**Example 1.** *Assume that the prospective data sources for sensor fusion are selected as follows:*

- *Face recognizer: This is attached to the corresponding physical instance.*
- *Collapse detector: This is attached to the corresponding physical instance.*
- *UAV camera: This is attached to the corresponding geographical area.*

The data sources Face Recognizer and Collapse Detector are attached to the instances of class Residence using the following queries:

- ```
  Q1 ATTACH_ENTRANCE WHERE
  physical_object == 'Residence' AND data_source == 'FaceRecognizer'
  ```

- Q2 ATTACH_BASEMENT WHERE
  physical_object == 'Residence' AND data_source == 'CollapseDedector'
- Q3 ATTACH_ENTRANCE WHERE
  physical_object == 'Residence' AND data_source == 'FaceRecognizer'
  AND physical_object.id == 'instance'
- Q4 ATTACH_ENTRANCE WHERE
  physical_object == 'Residence' AND data_source == 'FaceRecognizer'
  AND physical_object.robustness_factor '<3'
- Q5 ATTACH WHERE
  physical_object == 'GeographicRegion'
  AND physical_object.coordinates == 'coordinates'
  AND data_source == 'UAV-camera')

In Q1, 'ATTACH_ENTRANCE WHERE' is the query command, 'physical_object' denotes to one or more target objects of the query, and 'data_source' specifies the data source to be attached. This query selects all the instances of class Residence. Q2 is similar to Q1; however, as a data source, 'CollapseDedector' is attached. In Q3, with the help of the statement "physical_object.id == 'instance'" only one instance is selected. The word 'instance' here is used as a pseudo variable and must be replaced with a real instance defined in the simulation environment. Q4 selects all the instances of class Residence provided that their robustness factors are less than 3. Q5 selects a geographical region identified with the specified coordinate values and attaches a UAV camera as a data source.

The query specifications Q1 to Q5 are so-called 'write' queries. The system also supports 'read' queries which are embedded into certain menu items. These queries are called in the implementation of the menu items for analysis purposes, for example, to display the effectiveness, cost, and timing values of certain data fusion compositions. For brevity, these are not shown here.

## 6. Objectives and the Effects of Data Fusion

This section first defines the objectives of data fusion under five items. Second, based on the objectives, the combined effect of data fusion is presented.

### 6.1. Quality Objectives of Data Fusion

This section delineates the calculation of the individual effectiveness, cost, and timing values of the data sources that are introduced in the previous section.

In the Cambridge dictionary [59], effectiveness is defined as "achieving the result that you want". In our context, the effectiveness value is considered from the perspective of the following three objectives:

- **Objective 1:** The accuracy of estimating the effects of disasters is represented as a probabilistic variable. This value must be sufficiently high for a given purpose;
- **Objective 2:** The accuracy of estimating the horizontal coordinates of a (living) person after a disaster as a probabilistic variable. This value must be sufficiently high for a given purpose;
- **Objective 3:** The accuracy of estimating the vertical coordinates of a (living) person after a disaster as a probabilistic variable; this value must be sufficiently high for a given purpose.

For each data source, the effectiveness values can be taken from the related product catalogs and/or computed by experimentation. The effectiveness values must be computed per objective and assumed to be between 0 and 1, where 1 is the maximum possible effectiveness.

For each objective, different data sources can be selected and used. In case of adoption of multiple data sources for a given objective, the total effectiveness value is computed according to the data fusion formulas, which are given in Section 6.2.

The cost and timing values are considered by the objectives 4 and 5:

- **Objective 4:** The estimated cost value of a data fusion per element which is represented as a probabilistic distribution function. This value must not exceed the budgeting constraints;
- **Objective 5:** The estimated timing value of a data fusion per element which is represented as a probabilistic distribution function. This value must not exceed the deadline constraints.

In Appendix A, three tables are defined that contain the estimated parameters for the effectiveness, cost, and timing values of the prospective data sources.

*6.2. Calculating the Effect of Data Fusion*

The estimated effect of fusion of a set of data sources depends on the contribution of each data source if they are attached to the same element. The effect must be estimated according to one of the objectives defined in the previous section.

For determining the effectiveness values, depending on the characteristics of the geographical elements, prospective data sources, and the size and structure of the fusion, one may use different models of computation. Arithmetical formulas, statistical methods [19], Bayesian networks [20], Markov models, [60] etc., are typical examples. We observe, however, three determining factors: First, the model of computation must match the physical properties of the fusion. Second, it must be computationally efficient since these computations may need to be carried out many times especially during a synthesis process. Third, a library of alternative algorithms must be available if desired. This will give flexibility to the user in deciding the best algorithm for a particular case.

In this article, the unit of calculation is based on one of the element types within a geographical area such as a residence.

Basically, there are two cases:

1. **Effectiveness of fusion of data sources with no contribution to each other**. In this case, the effectiveness value of the data source with the highest value is considered. For example, in Appendix B Table A4, it is estimated that the data sources of UAV cameras, motion-based collapse detectors, seismic detectors, and cameras of the face recognizers have no contribution to a GPS tracker.
2. **Effectiveness of fusion of data sources with some contribution two each other**. The effectiveness value of the fusion must be computed.

In the following, we define three alternative formulas; Weak, Average, and Strong Contributions:

- **Weak contribution:** Per effectiveness objective, the selected $N$ data sources are ranked according to their effectiveness value from $1..N$, where 1 represents the data source with the highest effectiveness value and $N$ is the lowest. The effectiveness of data fusion of the element $k \in K$ which is denoted as $E_k$ with respect to objective $j \in J$ which is denoted as $O_j$ is calculated using the following formula:

$$EF(E_k, O_j) = e_1 + \sum_{n=2}^{N} \prod_{m=1}^{n-1} (1 - e_m)^2 \cdot e_n^2 \qquad (1)$$

where, $EF(E_k, O_j)$ is the estimated effectiveness value of the data fusion computed as a series of contributions of the ranked data sources, $e_1$ is the effectiveness value of the first data source in ranking, $e_m$ and $e_n$ are the effectiveness values of the $m$th and $n$th data sources in ranking, respectively.

- **Medium contribution:** This formula uses the same parameters as the Equation (1). The only difference is, in Equation (2), $(1 - e_m)$ the component of the formula is not squared.

$$EF(E_k, O_j) = e_1 + \sum_{n=2}^{N} \prod_{m=1}^{n-1} (1 - e_m) \cdot e_n^2 \qquad (2)$$

- **Strong contribution:** This formula uses the same parameters as the Equation (2). The only difference is, in Equation (3), the $e_n$ component of the formula is not squared.

$$EF(E_k, O_j) = e_1 + \sum_{n=2}^{N} \prod_{m=1}^{n-1} (1 - e_m) \cdot e_n \qquad (3)$$

The effects of the three formulas are illustrated in the Figure 4. Here, the green, orange, and blue lines display the weak, medium, and strong contributions, respectively. This figure is drawn with the assumption that every additional data source has the effectiveness value 0.5. It can be seen that green contribution has an upper bound of approximately 0.6.

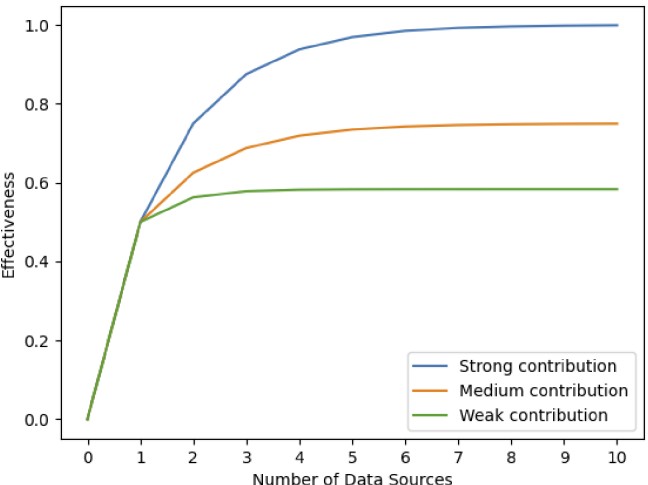

**Figure 4.** Effectiveness comparison between weakly, averagely, strongly related data sources.

**Example 2.** *Let us assume that the following data sources are to be considered for the residence R:*

- *A UAV-camera is dispatched to the geographical area.*
- *There is a base station in the geographical area.*
- *One collapse detector is installed at the ground floor.*
- *There exists a registration database.*
- *There is a face recognizer at the entrance of the building.*
- *The camera of the face recognizer is used also as a collapse detector.*

Based on Appendices A and B , the objective values are computed as follows:

**Objective 1 (effectiveness)**: To estimate the effectiveness value for this objective, using the Tables A1 and A2 from Appendix A and based on the mean values of the distribution functions, the data sources under consideration are ranked:

1. Collapse detector (0.7).
2. UAV-camera (0.7).
3. The camera of the face recognizer (0.65).
4. Base station (0).
5. Registration database (0).
6. Face recognizer (0).

The data sources 4, 5, and 6 do not contribute to the considered effectiveness objective.

The data sources 1, 2, and 3 contribute strongly to each other.

The estimated effectiveness value of the residence $R$ with respect to objective 1 can be computed using Equation (3):

$$EF(E_R, O_1) = 0.7 + (1 - 0.7) \times (0.7) + (1 - (1 - 0.7) \times 0.7) \times (0.65) = 0.9685$$

**Objective 2 (effectiveness):** To calculate the estimated effectiveness value of this objective, using the Tables A1 and A2 and based on the mean values of the distribution functions, the data sources under consideration are ranked:

1. Face recognizer (0.75).
2. Registration database (0.55).
3. Collapse detector (0).
4. UAV-camera (0).
5. The camera of the face recognizer (0).
6. Base station (0).

The data sources 3, 4, 5, and 6 do not contribute to the considered effectiveness objective. The data sources 1 and 2 strongly contribute to each other.

The estimated effectiveness value of the residence $R$ with respect to objective 2 can be computed using Equation (3):

$$EF(E_R, O_2) = 0.75 + (1 - 0.75) \times (0.55) = 0.8875$$

**Objective 3 (effectiveness):** This estimated effectiveness value of objective 3 is the same as the one of objective 2.

**Objective 4 (total cost value):** The total cost value is estimated by adding up the cost values of the attached data sources for each element. Assuming the following five data sources are utilized:

1. Face recognizer: 12.5 K .
2. Collapse detector: 5.5 K.
3. UAV-camera: This cost value is not included.
4. Base station: This cost value is not included.
5. Registration database: This cost value is not included.

Total cost of ($R$) = 18 K

**Objective 5 (total timing value):** The total timing value is estimated as follows: Assuming the following five data sources are utilized, the data sources are ranked according to their timing values:

1. Face recognizer: 0.55.
2. Collapse detector: 0.55.
3. Registration database: 0.55.
4. Base station: 10.5.
5. UAV-camera: 8.5 K.

Compared to the UAV, the timing values of the other data sources are negligible. As depicted in the following Figure 5, the estimated timing value, for example, with respect to objective 1, is computed in two steps:

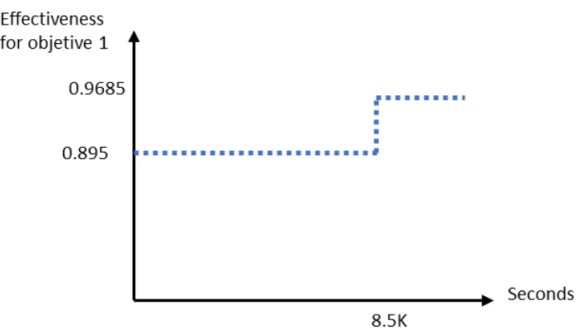

**Figure 5.** Estimated effectiveness value of the example with respect to objective 1.

## 7. Algorithms for Synthesizing the Optimal Data Fusion Configuration

The proposed approach in this article consists of two phases:

1. Forming a model of a geographical area with a set of data sources attached.
2. Calculating the effectiveness, cost, and timing properties of the model.

In the previous sections, forming a model is exemplified as a set of manual actions carried out by the analyst. Although manual steps may be feasible in small scales, it may become intractable if the model becomes too large. This article proposes two algorithms for automated synthesis: *Best-fit* and *Optimal-fit* [41].

### 7.1. Best-Fit

This is a 'greedy' algorithm that builds up a search space of possible element/data-source compositions step by step. At each step, the data source which has the best effectiveness value is kept. The other alternatives are discarded. This process terminates until all prospective data sources are considered and/or the cost and timing constraints are violated. According to this approach, the data sources which are selected until the termination phase give the optimal composition for fusion.

The Algorithm 1 is implemented by the function *CALCULATE_BEST_FIT*. The parameters used in this algorithm are *instance, time_limit*, and *cost_limit*, which represent an instance of a physical object, timing and cost constraints, respectively. Lines (2) to (4) are used to create the initial setting of the search tree, which represents the possible alternatives of the data sources attached to a given instance. Here, *instance* represents a geographical element such as an instance of class Residence. Selecting the best data source at each step is realized by calling on the function *BEST_APPLICABLE_SOURCE* in line (5). This function returns the best data source at a given step. This data source is appended to the design tree as a mode. The total cost and timing values are calculated and updated. This process continues until the time or cost limits are violated or all data sources are considered (line 12).

The function *BEST_APPLICABLE_SOURCE* searches for all possible data sources for the given context of the instance and the related tree structure. It accepts the parameters *ds-_tree, instance, time_limit*, and *cost_limit* and returns the data source that has the maximum contribution to the effectiveness value. The function *CALCULATE_EFFECTIVENESS* implements the formulas given in Section 6.2.

**Example 3.** *Let us assume that the best data fusion composition must be found for an instance of class Residence. The following characteristics are assumed:*

- *The cost limit is 15 K in unit of currency.*
- *The time limit is 3.6 K in seconds.*

---

**Algorithm 1** CALCULATE_BEST_FIT(instance, time_limit, cost_limit) returns list of data sources

---

1: **function** CALCULATE_BEST_FIT(instance, time_limit, cost_limit)
2:     *time* ← 0
3:     *cost* ← 0
4:     *ds_tree* ← empty tree
5:     *time*, *cost* ← *ds_tree.add_root*(*BEST_APPLICABLE_SOURCE*(*ds_tree*, *instance*, *time_limit*, *cost_limit*))
6:     **while** constraints are met **do**
7:         *t*, *c* ← *ds_tree.add_child*(*prev_element*, *BEST_APPLICABLE_SOURCE*(*ds_tree*, *instance*, *time_limit*, *cost_limit*))
8:         **if** new data_source attached **then**
9:             *time* += *t*
10:            *cost* += *c*
11:         **else**
12:            **break**
13:         **end if**
14:     **end while**
15:     **return** *ds_tree*
16: **end function**

17: **function** BEST_APPLICABLE_SOURCE(ds_tree, instance, time_limit, cost_limit)
18:     *best_data_source* ← empty data_source object
19:     *found* ← false
20:     *data_sources* ← available data sources for the instance, length *N*
21:     *effectiveness_values* ← empty list with length *N* used for comparison
22:     **for** index from 0 to *N* **do**
23:         *effectiveness*, *time*, *cost* ← *CALCULATE_EFFECTIVENESS*(*ds_tree*, *data_sources*[*N*])
24:         **if** *time* < *time_limit* and *cost* < *cost_limit* **then**
25:            *effectiveness_values*[*index*] ← *effectiveness*
26:            *found* ← true
27:         **end if**
28:     **end for**
29:     **if** found **then**
30:         *best_data_source* ← *data_sources*[argmax(*effectiveness_values*)]
31:     **end if**
32:     **return** *best_data_source*
33: **end function**

---

The effectiveness values of the available data sources are estimated as follows:

1. Smart home ($e = 0.717$);
2. Face recognizer ($e = 0.5$);
3. Collapse detector ($e = 0.233$);
4. Camera of face recognizer($e = 0.22$);
5. Seismic detector ($e = 0.133$);
6. People counter ($e = 0.133$).

For brevity, the cost and timing values of these data sources are not shown here. In the first iteration, smart home is selected since it has the highest estimated effectiveness value. The total cost is calculated as 2.5 K units of currency and the total time value is 0.1 s. In the second iteration, the contributions of the remaining data sources are calculated. The face recognizer is selected because it has the highest contribution value: $(0.717 + (1 - 0.717) \times 0.5 = 0.8585)$.

The total cost is estimated as 15 K and the total time as 0.655 s. The algorithm reaches its cost limit and therefore terminates.

In this example, we assume that only one data source of the same type can be attached to an residence. For illustration purposes, to explain the algorithm, a relatively simple example is chosen. In a real case, however, there may be many geographical elements to be considered with a large number of alternatives and variations of data sources.

The running time of the algorithm can be calculated as follows: If the number of data sources in the output is $N$, and the number of available data sources is $M$, then the running time can be computed by (4), where t represents the running time in each allocation step (corresponds to the function $BEST\_APPLICABLE\_SOURCE$).

$$T_{best} = t \times (\sum_{k=1}^{M} M - k + 1)$$
$$T_{best} = t \times N \times (M - N + \frac{1}{2})$$

(4)

### 7.2. Optimal-Fit

Similar to the previous one, this Algorithm 2 builds up a search space of possible element/data-source compositions step by step. Contrary to the previous one, at each step, for all data source compositions, the effectiveness, time, and cost values are calculated and the alternatives with higher effectiveness values are appended to the search tree. This process terminates if all data sources are considered or the timing and cost constraints are violated. The branch with the highest effectiveness value is considered as the optimal data source composition.

The Algorithm 2 is implemented by the function $CALCULATE\_OPTIMAL\_FIT$. The parameters used in this algorithm, same as the previous Algorithm 1, are *instance*, *time_limit*, and *cost_limit*, which represent an instance of a physical object, timing and cost constraints, respectively. In line (2), the function $OPTIMAL\_APPLICABLE\_SOURCES$ is called, which returns a tree whose leaves refer to a set of data source alternatives that have the maximum contribution to the effectiveness on their path with respect to timing and cost constraints.

In the specification of the function $OPTIMAL\_APPLICABLE\_SOURCES$, two functions play an important role: The function $GET\_MAX\_N$ implements the formulas defined in Section 6.2. For each leaf node, the function $GETPATH$ searches for the paths to the root node and calculates the total effectiveness, cost, and timing values of the branch. According to this approach, the path that has the highest effectiveness value is selected as the representation of the optimal data fusion.

The running time of the algorithm can be calculated as follows: If the tree length in the output is $\mathcal{L}$, and the number of available data sensors is $M$, and the number of children nodes created is $C = 3$, then the running time is lower than the value showed in (5). Here, t represents the running time in each allocation step (corresponds to the function $OPTIMAL\_AVAILABLE\_SOURCES$). An exact running time cannot be calculated, because the level of leaf nodes is variable depending on the cost limit.

$$T_{optimal} \leq t \times [M + C^{\mathcal{L}-1} \sum_{k=1}^{C} M - k - \mathcal{L} + 2], \ \mathcal{L} > 1$$
$$T_{optimal} \leq t \times [M + C^{\mathcal{L}}(M - \mathcal{L} + \frac{C+5}{2})], \ \mathcal{L} > 1$$

(5)

---

**Algorithm 2** CALCULATE_OPTIMAL_FIT(instance, time_limit, cost_limit) returns list of data sources

---

 1: **function** CALCULATE_OPTIMAL_FIT(instance, time_limit, cost_limit)
 2:     *candidate_trees* ← *OPTIMAL_APPLICABLE_SOURCES(empty_tree, instance,*
    *time_limit, cost_limit)*
 3:     **while** constraints are met **do**
 4:         *candidate_trees* ← *OPTIMAL_APPLICABLE_SOURCES(candidate_trees,*
    *instance, time_limit, cost_limit)*
 5:         **if** each leaf of *candidate_trees* is *end_leaf* **then**
 6:             **break**
 7:         **end if**
 8:     **end while**
 9:     *best_path* ← *empty_tree*
10:     **for** each leaf in *candidate_trees* **do**
11:         *path, effectiveness* ← *GETPATH(candidate_trees, leaf)*     ▷ returns subtree
12:         **if** *effectiveness* > *best_path.effectiveness* **then**
13:             *best_path* ← *path*
14:         **end if**
15:     **end for**
16:     **return** *best_path*
17: **end function**

18: **function** OPTIMAL_AVAILABLE_SOURCES(*ds_trees, instance, time_limit, cost_limit*)
19:     *available_data_sources* ← GET_AVAILABLE_SOURCES(*ds_trees, instance*)   ▷ 2D
    list that returns available data sources for each subtree
20:     **for all** *leaf* **in** *ds_trees* **do**
21:         *expanded* ← false
22:         *path, effectiveness* ← GETPATH(*ds_trees, leaf*)
23:         *selected_data_sources* ← GET_MAX_N(*path, available_data_sources*, 3)   ▷ Get
    *N* maximum effectiveness values
24:         **for all** *data_source* **in** *selected_data_sources* **do**
25:             *subtree.addchild(data_source)*
26:             *expanded* ← true
27:         **end for**
28:         **if** not *expanded* **then**
29:             *subtree.leaf.type* ← *end_node*
30:         **end if**
31:     **end for**
32:     **return** *ds_trees*
33: **end function**

---

**Example 4.** *Consider the following data sources where the optimal data fusion option must be determined:*

- *The cost limit is 15 K in unit of currency.*
- *The time limit is 3.6 K in seconds.*

The effectiveness values of the available data sources are estimated as follows:

1. GPS trackers ($e = 0.433$);
2. Mobile phone apps ($e = 0.433$);
3. Base station ($e = 0.367$);
4. Registration database ($e = 0.333$);
5. UAV-camera ($e = 0.233$).

In this example, we only consider the effectiveness values. In the first iteration step, the data sources with highest effectiveness values are (GPS trackers, mobile phone apps, and base stations). In the second iteration, the best three data sources must be found for

each data source that has been selected in the previous step. The result of the second iteration is given below:

- GPS Trackers → Mobile Phones, Base Stations, Registration Database;
- Mobile Phone apps → GPS Trackers, Base Stations, Registration Database;
- Base Stations → GPS Trackers, Mobile Phones, Registration Database.

Here, the sign → denotes to appended sub-nodes. Assume that the algorithm is now terminated. The final step is to determine the best path from the nine paths. There are two paths with the same value: ($e = 0.679$). The data sources in each path are: {GPS Trackers, Mobile Phone apps} and {Mobile Phone apps, GPS Trackers}.

## 8. Generalization of the Analysis and Synthesis Approach to UAS-Based Data Fusion

In Section 2.4, it was stated that the concept of AAM and its implementation as UAS can bring new opportunities for improving situation awareness in disaster/earthquake management.

A UAV, together with many other system components such as ground-based stations and data sources of many kinds, provides much better coverage. They can form all together so-called systems of systems, in which swarm of UAVs or even swarm UASs can dynamically exchange data and information, and cooperate with each other for a common set of goals. In addition to generic ones, UASs can also be designed for specific missions such as disaster management. They may also share the heavy load such as radars and precision optical devices.

Such a geography-based system consisting of static and dynamic parts can offer effective decision making. In addition, life data can be provided to emergency control centers. UASs, in short, offer the perspective of creating a more dynamic and self-adapting and responsive system architecture for the purpose of dealing with the unforeseen disaster conditions.

In the following, we therefore elaborate on the possibilities of extending our analysis and design framework for supporting UASs.

AAM-based data fusion systems can be introduced to the analysis and synthesis framework in the following way:

1. UAV must be introduced as an element of a geographical model. If necessary, a new class must be introduced in CityGML.
2. A set of queries must be defined for UAVs so that data sources can be attached for fusion. In this case, an instance of a UAV can function as a data source and and as a fusion node (element of a geographical area).
3. In addition to cost and timing values, a new quality attribute *weight* must be introduced.
4. The analysis and synthesis algorithms must take care of this new attribute as well.
5. UAVs may cooperate together during their mission by sharing some of their tasks.
6. To analye and synthesize models with cooperating UAVss, the computation of the efficiency values must take care of a group of elements. In addition, time-dependent properties of UAVs must be taken into account. The analysis and synthesis algorithms must be defined accordingly, possibly by using network-based evaluation models.

The steps described in this section illustrate how the framework can be adapted to cope with substantial change demands. These are our future plans for research.

## 9. Discussion

This section elaborates on the following conditions which may violate the assumptions made in this article:

- **Data sets and incorrect assumptions of the effectiveness, cost, and timing values of data sources:** The data sets used in the examples of this article are based on the characteristics of the data sources in Appendices A and B. Each value is expressed as a probabilistic variable of uniform distribution within a certain range.

  Although carefully defined, these values may differ considerably from some of the commercially available data sources in the market. Moreover, with the advancement

of technology, new products are introduced frequently. It is therefore advisable to consider concrete products instead of their abstract representations. In case of adoption of concrete products, the accuracy of the estimations can be improved by consulting to the catalogues, and if necessary, by carrying out dedicated experiments. If sufficient data are available, machine learning techniques can be adopted to improve these values as well. Nevertheless, the methods and techniques introduced in this article do not depend on the data values presented in Appendices A and B; the data values are used for illustration purposes and in the examples only.

- **Inaccurate data fusion formulas:** The data fusion formulas presented in Section 6.2 are based on the following assumptions: (a) The relevancy factor of a data fusion for the objectives 1 to 3 is a probabilistic variable defined in the range of 0 to 1. (b) Attaching a new data source cannot degrade the effectiveness factors of the already attached data sources. (c) If a newly added data source contributes to the considered objective, the effectiveness function is a monotonously increasing function asymptotically approaching 1 or a value less than 1. We consider these assumptions reasonable.

  The formulas used for weak, medium, and strong contributions can be adapted to the needs, or new formulas can be introduced as plug-ins. A limitation to this approach is that data fusion is assumed to be realized at a geographical element only. In Section 8, more general fusion possibilities are discussed.

  New formulas can be defined in various ways, as long as they do not violate the assumptions made. The contribution factors of the data sources to each other as presented in Appendix B can be improved by experimentation. In addition, if sufficient data are available, machine learning techniques can be adopted to improve these values.

- **Extensibility of the framework:** Due to evolution of the needs and technologies, it may be necessary to introduce new elements and/or data sources. The model-based architectural style as described in Section 5 provides flexibility. For example, to introduce a new geographical element, the following actions must be carried out: (1) A new class representing the element must be introduced in the GIS model, possibly by subclassing the existing classes. (2) The attributes of the instances of the class must be initialized including the command objects for the relevant queries. The menu items of the user interface can be automatically generated from the command objects by using the Command design pattern [44]. If a new data source is to be introduced, the following steps must be carried out: (1) The feature-model must be edited to introduce the new feature, which represents the new data source. (2) A new set of command objects must be added to the relevant element instances to enable attaching of the new data source, if necessary. (3) The tables used in computing the effectiveness, cost, and timing values must be updated. Changes to computations can be introduced as plug-ins.

- **Complexity of automatic synthesis of data fusion**: If the number of possible data sources which can be attached to a selected element is large and if this element offers a large number of alternatives for data sources, the search space of the optimization algorithm can be too large to handle.

  In this article, we adopt a heuristic rule based on the following: First, data sources are ranked according to their effectiveness values. Second, the search space is formed by starting from the alternatives with the highest effectiveness values. Gradually, other alternatives are considered according to their ranking order. The process continues until the whole search space is constructed or the cost and/or timing constraints are violated. It is also possible to limit the size of the search space while constructing it. The heuristic rule reduces the state space considerably. This algorithm may not find the optimal fusion if many data sources with fewer effectiveness values give in total a better result than a few but more effective data sources. However, in practice due to physical restrictions, it may be impractical to attach too many data sources at a given

geographical area even if their total effectiveness value is high. The adopted heuristic rule is therefore considered preferable for most cases.

The algorithms presented in this article adopts a single objective optimization strategy, meaning that the quality attribute effectiveness is the main objective of the search for the optimal solution. The other attributes, cost and timing values, are the restricting constraints. One can also adopt multi-objective-based optimization algorithms, such as Pareto optimization [61], to consider all relevant quality factors. From the perspective of this article, the effectiveness of earthquake damage detection is the main objective and the other two attributes are only taken into account as limiting constraints.

## 10. Future Work

The article presents part of our ongoing activities on disaster management. We are currently extending the proposed approach in various ways. First, we are setting up a laboratory to measure the actual parameters of commercially available data sources. By this way, the data values will be determined more precisely for a given set of products. In our current approach, data fusion is realized at certain predetermined physical locations. This approach will be extended to UAS-based data Ffsion. In this way, it will be possible to analyze and synthesize the models with a cooperating swarm of UAVs. Finally, the analysis and synthesis algorithms will be investigated for this dynamically changing topology of data-fusion systems.

## 11. Results and Conclusions

After the occurrence of a disaster such as an earthquake, emergency control centers need to accurately and promptly determine the effect of the disaster so that aid operations can be carried out effectively. To this aim, a data fusion system consisting of multiple data sources must be designed and installed before a disaster occurs. Unfortunately, designing a cost-effective system for this purpose is a challenging task. Firstly, the kinds of possible data sources can be too large. Secondly, each data source may have its relative advantages and shortcomings. Moreover, in case of data fusion, mutual effects of data sources on each other must be considered. Finally, data sources may need to be installed in a large geographical area.

We will now elaborate on this article from the perspective of the contributions/novelties by referring to the research questions formulated in Section 3.

A novel model-based analysis and synthesis framework is introduced to address the research questions 'What is a suitable architectural style of the desired model-based framework?' and 'How can this architecture be extended to deal with the (future) UAS?'. In Section 8, we have outlined an approach for extending the proposed framework so that it can be utilized in designing UASs.

Through an extensive domain analysis work and by organizing the identified domain concepts under a feature diagram, a new domain model is proposed to address the research question 'How to define a domain model of the data sources suitable for detecting the effects of earthquakes?'. Furthermore, a detailed specification of the relevant data sources from the perspective of data fusion is presented in Appendices A and B.

A novel set of extensible object-oriented models and queries is introduced to give an answer to the research question 'How to compute the combined effectiveness, cost and timing values as a result of data fusion?'. These can be used for simulating the prospective data fusion alternatives. In addition, formulas are defined to determine whether the quality objectives are fulfilled or not.

Novel automatic synthesis algorithms for data fusion are proposed to address the research question 'What kind of algorithms can be defined for automated synthesis of the optimal data fusion?'.

The utility of the proposed approach is illustrated by a set of examples.

Overall, the proposed approach may eliminate the risks of designing and installing irrelevant or less effective data fusion systems. We consider that the research questions

presented in Section 3 are adequately addressed within the limits of the assumptions, as discussed in the previous section.

**Author Contributions:** M.A. had the original idea for this work. He has elaborated and written most of the sections. Students H.S. and M.A.E. and V.V.d.C. have concentrated their efforts on some background-related points, elaboration of the object-oriented model (Section 5), implementing the related code, and reviewing the paper. All authors have read and agreed on the published version of the manuscript.

**Funding:** The TÜBİTAK BİDEB program 2232 International Fellowship for Outstanding Researchers.

**Data Availability Statement:** No new data were created or analyzed in this study. Data sharing is not applicable to this article.

**Acknowledgments:** B. Akkus, as a student, contributed to an earlier version of the domain model. This work has been generously supported by the TÜBİTAK BİDEB program 2232 International Fellowship for Outstanding Researchers.

**Conflicts of Interest:** The authors declare no conflict of interest.

## Appendix A. Estimated Parameters of Data Sources

None of the algorithms of this article are dependent on the data values presented in this appendix. Data values are estimated based on the characteristics of some selected products available in the market. Since technology is rapidly evolving, detection accuracy, price, and response time of the presented data sources may vary in due time. When applying the method and techniques presented in this article, the data values must be obtained from the catalogs of the data sources to be purchased. Moreover, for brevity, not all the possible data source types are represented in Appendices A and B.

*Appendix A.1. Data Sources Attached to Geographical Areas*

Table A1 is used to document the estimated characteristics of data sources that can be attached to geographical areas. Here, the first column lists the considered data sources. The remaining columns list the individual characteristics of these data sources with respect to the five objectives presented in Section 4. The objectives are abbreviated as Obj$<$i$>$ where the index i is defined between 1 to 5. The min and max columns refer to the estimated range of the characteristics per objective. From Obj1 to Obj3, the values in cells indicate the estimated effectiveness values of data sources. These are expressed as probabilistic variables, where the corresponding min and max columns indicate their ranges. The columns Obj4 and Obj5 refer to the cost and timing characteristics of the data sources, respectively. These are expressed as probability distribution functions, where the corresponding min and max columns refer to their ranges.

**Table A1.** The Estimated Parameters of a Selected Set of Relevant Data Sources

| Data Source | Obj1 * min. | Obj1 * max. | Obj2 * min. | Obj2 * max. | Obj3 * min. | Obj3 * max. | Obj4 ^ min. | Obj4 ^ max. | Obj5 + min. | Obj5 + max. |
|---|---|---|---|---|---|---|---|---|---|---|
| UAV (Camera) | 0.5 [&] | 0.9 [&] | 0 | 0 | 0 | 0 | - | - | 1.8 K | 18 K |
| GPS trackers | 0 | 0 | 0.4 | 0.9 | 0.4 | 0.9 | 8 K | 20 K | 0.01 | 0.5 |
| Mobile phones | 0 | 0 | 0.4 | 0.9 | 0.4 | 0.9 | - | - | 0.01 | 0.5 |
| Base stations | 0 | 0 | 0.3 | 0.8 | 0.3 | 0.8 | - | - | 1 | 20 |
| Registration database | 0 | 0 | 0.3 | 0.8 | 0.1 ** | 0.8 ** | - | - | 0.1 | 1 |

* Probability of effectiveness: uniform distribution. ^ Probability of cost value: uniform distribution in units of currency. The cost of UAVs, Base stations, and Registration databases are neglected in our calculations per physical object. Due to the high cost values of airborne data sources, they must be budgeted separately. + Probability of timing value: uniform distribution in units of seconds. ** These values depend on day or night time, and on holiday periods. & Largely dependent on weather conditions.

*Appendix A.2. Data Sources Attached to Physical Objects*

Table A2 is used to document the estimated characteristics of data sources that can be attached to physical objects. The definitions of the columns and rows are similar to the ones of Table A1.

**Table A2.** The Estimated Parameters of a Selected Set of Relevant Data Sources

| Data Source | Obj1 * min. | Obj1 * max. | Obj2 * min. | Obj2 * max. | Obj3 * min. | Obj3 * max. | Obj4 ^ min | Obj4 ^ max. | Obj5 + min. | Obj5 + max. |
|---|---|---|---|---|---|---|---|---|---|---|
| Motion-based collapse detector | 0.6 | 0.8 | 0 | 0 | 0 | 0 | 1 K | 10 K | 0.1 | 1 |
| Seismic detector | 0.2 ** | 0.6 ** | 0 | 0 | 0 | 0 | 500 | 2 K | 0.01 | 0.5 |
| Smart-home detectors | 0.8 ^^ | 0.9 ^^ | 0.4 | 0.9 | 0.4 | 0.9 | 1 | 4 K | 0.01 | 0.2 |
| Face recognizer | 0 | 0 | 0.6 ++ | 0.9 ++ | 0.6 ++ | 0.9 ++ | 5 K | 20 K | 0.1 | 1 |
| Camera of the face recognizer | 0.5 | 0.8 | 0 | 0 | 0 | 0 | 0 | 0 | 0.1 | 1 |
| People counter & | 0 | 0 | 0.1 | 0.3 | 0.1 | 0.3 | 500 | 1 K | 1 | 10 |

* Probability of effectiveness: uniform distribution. ^ Probability of cost value: uniform distribution in units of currency. + Probability of timing value: uniform distribution in units of seconds. ** If the earthquake resistance factor of physical elements can be estimated accurately. Otherwise, we assume the values of 0.1 to 0.4. ^^ Except the status of collapse, which is assumed to be zero. ++ If the disaster occurs after a certain time of detection, the detected persons can be assumed outside of the vicinity or at their residences. & Can only estimate the number of persons in the building.

*Appendix A.3. Data Sources Transported to the Locations of Disaster Areas*

Table A3 is used to document the estimated characteristics of data sources that can be transported to disaster areas. The definitions of the columns and rows are similar to the ones of Table A1.

**Table A3.** The Estimated Parameters of a Selected Set of Relevant Data Sources

| Data Source | Obj1 * min. | Obj1 * max. | Obj2 * min. | Obj2 * max. | Obj3 * min. | Obj3 * max. | Obj4 ^ min | Obj4 ^ max. | Obj5 + min. | Obj5 + max. |
|---|---|---|---|---|---|---|---|---|---|---|
| Microphone | 0 | 0 | 0.05 | 0.1 | 0.1 | 0.4 | 1 K | 10 K | 600 | 3.6 K |
| Carbon dioxide meter | 0 | 0 | 0.1 | 0.4 | 0.05 | 0.1 | 500 | 2 K | 600 | 3.6 K |
| Microwave radar | 0 | 0 | 0.8 | 0.9 | 0.8 | 0.9 | 2 M | 4 M | 600 | 3.6 K |

* Probability of effectiveness: uniform distribution. ^ Probability of cost value: uniform distribution in units of currency. The cost of microwave radar is neglected in our calculations per physical objects. Due to their high cost values, they must be budgeted separately. + Probability of timing value: uniform distribution in units of seconds. The transportation time of these data sources can be considerably high. Further, for microphones and carbon dioxide meters, it may be necessary to drill holes, which also takes extra time.

## Appendix B. Estimated Parameters of the Effectiveness of Fusion of Multiple Sources

Table A4 is used to document the estimated characteristics of the mutual effects of data sources to each other in case of data fusion. The first column and row list the considered data sources. The first row refers to the data sources symbolically, expressed as (a) to (n). These symbols refer to the data sources listed in the first column. For example, (a) refers to UAV cameras. The number "0" in a cell indicates that there is no contribution of the related data sources to each other. The values "1" and "2" indicate weak and strong contributions, respectively. For brevity, medium contribution values are not shown. The semantic meanings of the degree of contributions are defined by the formulas in Section 6.2.

**Table A4.** Estimated parameters of selected instances of data sources, transported to the locations of disasters.

|  | (a) | (b) | (c) | (d) | (e) | (f) | (g) | (h) | (i) | (j) | (k) | (l) | (m) | (n) |
|---|---|---|---|---|---|---|---|---|---|---|---|---|---|---|
| UAV-camera (a) | 1 + | 0 | 0 | 0 | 0 | 2 | 1 | 2 ^ | 0 | 2 | 0 | 0 | 0 | 0 |
| GPS trackers (b) | 0 | 0 | 2 | 2 | 2 * | 0 | 0 | 2 * | 2 * | 0 | 1 | 1 | 2 | 2 |
| Mobile phones (c) | 0 | 2 | 0 | 2 | 2 * | 0 | 0 | 2 * | 2 * | 0 | 1 | 1 | 2 | 2 |
| Base stations (d) | 0 | 2 | 2 | 1 | 2 | 0 | 0 | 2 * | 2 * | 0 | 1 | 1 | 2 | 2 |
| Registration database (e) | 0 | 2 * | 2 * | 2 | 0 | 0 | 0 | 0 | 2 * | 0 | 0 | 1 | 2 | 2 |
| Motion-based collapse detector (f) | 2 | 0 | 0 | 0 | 0 | 2 | 2 | 0 | 0 | 2 | 0 | 0 | 0 | 0 |
| Seismic detector (g) | 1 | 0 | 0 | 0 | 0 | 2 | 0 | 0 | 0 | 2 | 0 | 0 | 0 | 0 |
| Smart home detectors (h) | 2 ^ | 2 * | 2 * | 2 * | 0 | 0 | 0 | 2 | 2 * | 0 | 0 | 0 | 0 | 0 |
| Face recognizer (i) | 0 | 2 * | 2 * | 2 * | 2 * | 0 | 0 | 2 * | 0 | 0 | 0 | 2 | 2 | 2 |
| Camera of the face recognizer (j) | 2 | 0 | 0 | 0 | 0 | 2 | 2 | 0 | 0 | 0 | 1 | 0 | 0 | 0 |
| People counter (k) | 0 | 1 | 1 | 1 | 0 | 0 | 0 | 0 | 0 | 1 | 0 | 0 | 0 | 0 |
| Microphone (l) | 0 | 1 | 1 | 1 | 1 | 0 | 0 | 0 | 2 | 0 | 0 | 2 | 2 | 2 |
| Carbon dioxide meter (m) | 0 | 2 | 2 | 2 | 2 | 0 | 0 | 0 | 2 | 0 | 0 | 2 | 2 | 2 |
| Microwave radar (n) | 0 | 2 | 2 | 2 | 2 | 0 | 0 | 0 | 2 | 0 | 0 | 2 | 2 | 2 |

0 Data sources with no contribution to each other. 1 Data sources with weak contribution to each other. 2 Data sources with strong contribution to each other. + If additional UAV-camera has a surveillance mission in a better weather condition. ^ Disasters, other collapse conditions; we assume that smart home detectors do not detect collapses. * If a person is detected and his/her identity is known.

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
