# Peer review of "Data Fusion Analysis and Synthesis Framework for Improving Disaster Situation Awareness"

_drones, doi:10.3390/drones7090565_

Round 1

Reviewer 1 Report

The manuscript entitled "DATA FUSION ANALYSIS AND SYNTHESIS FRAMEWORK FOR IMPROVING DISASTER SITUATION AWARENESS" has been investigated in detail. The topic addressed in the manuscript is potentially interesting and the manuscript contains some practical meanings, however, there are some issues which should be addressed by the authors:

ü  Provide highlights and graphical abstract per journal instructions.

ü  The authors need to emphasize their contributions/novelties in the revision. In the current version, the authors did not discuss their contributions in detail.

ü  The authors should carefully proofread this paper and correct all the typos in the revision. In the current version, there are still some typos/grammar errors.

ü  Did the authors introduce the details of the dataset?

ü  The references are old, use newer references.

ü  The authors should discuss future work plans.

ü  The presentation of this paper needs some improvement. For example, some parts have unnecessary empty spaces. The authors should remove these unnecessary empty spaces in the revision.

ü  Could the authors report the running time of the proposed algorithm? In this way, we can justify whether this algorithm can be applied to large-scale dataset.

ü  The authors should address the following limitations. The first limitation is that, the related works should be grouped into two or three subsections. In the current version, the authors all merged them together.

ü  The proposed algorithm still can be improved if the ideas in the following papers are explored, i.e., "Simultaneous localization and mapping (slam) and data fusion in unmanned aerial vehicles: Recent advances and challenges", "Semantic frameworks to enhance situation awareness for defence and security applications", "Deep learning-based face detection and recognition on drones", and "Aerial orthoimage generation for UAV remote sensing". The authors are encouraged to discuss them in the revision.

The manuscript entitled "DATA FUSION ANALYSIS AND SYNTHESIS FRAMEWORK FOR IMPROVING DISASTER SITUATION AWARENESS" has been investigated in detail. The topic addressed in the manuscript is potentially interesting and the manuscript contains some practical meanings, however, there are some issues which should be addressed by the authors:

ü  Provide highlights and graphical abstract per journal instructions.

ü  The authors need to emphasize their contributions/novelties in the revision. In the current version, the authors did not discuss their contributions in detail.

ü  The authors should carefully proofread this paper and correct all the typos in the revision. In the current version, there are still some typos/grammar errors.

ü  Did the authors introduce the details of the dataset?

ü  The references are old, use newer references.

ü  The authors should discuss future work plans.

ü  The presentation of this paper needs some improvement. For example, some parts have unnecessary empty spaces. The authors should remove these unnecessary empty spaces in the revision.

ü  Could the authors report the running time of the proposed algorithm? In this way, we can justify whether this algorithm can be applied to large-scale dataset.

ü  The authors should address the following limitations. The first limitation is that, the related works should be grouped into two or three subsections. In the current version, the authors all merged them together.

ü  The proposed algorithm still can be improved if the ideas in the following papers are explored, i.e., "Simultaneous localization and mapping (slam) and data fusion in unmanned aerial vehicles: Recent advances and challenges", "Semantic frameworks to enhance situation awareness for defence and security applications", "Deep learning-based face detection and recognition on drones", and "Aerial orthoimage generation for UAV remote sensing". The authors are encouraged to discuss them in the revision.

Author Response

[Drones] Manuscript ID: drones-2458493

Dear Editor(s)

Thank you very much for giving a detailed review of our submitted paper for the Drones special issue. We have considered the remarks seriously and worked on them carefully. We have improved our paper accordingly and realized the following improvements:

  • Remark: The authors need to emphasize their contributions/novelties in the revision. In the current version, the authors did not discuss their contributions in detail.

Our improvement-01:

We have emphasized contributions/novelties in the last revision as follows:

  1. Ä°n introduction, the following paragraph is written to emphasize the contributions/novelties:

The contributions/novelties of this article are as follows. … (Please refer to the article).

  1. In Section 11, contributions/novelties are emphasized by writing the following paragraph:

We will now elaborate on this article from the perspective of the contributions/novelties by referring to the research questions formulated in Section 3. …. (Please refer to the article).

  • Remark: The authors should carefully proofread this paper and correct all the typos in the revision. In the current version, there are still some typos/grammar errors.

Our improvement-02: We have thoroughly checked the article for typos/grammar errors according to the US English spelling rules. Thank you for emphasizing this issue.

  • Remark: Did the authors introduce the details of the dataset?

Our improvement-03: Our data references are the characteristics of the data sources, which are presented in Appendix. In addition, we have added the following item in Section 9:

Data sets and incorrect assumptions of the effectiveness, cost, and timing values of data sources: … (Please refer to the article).

  • Remark: The references are old, use newer references.

Our improvement-04: We have added more up to date references into Section 2.

  • Remark: The authors should discuss future work plans.

Our improvement-05: We have added a new section (section 10) Future plans for this purpose.

  • Remark: The presentation of this paper needs some improvement. For example, some parts have unnecessary empty spaces. The authors should remove these unnecessary empty spaces in the revision.

Our improvement-06: We have thoroughly scanned the article and removed the empty spaces unless readability of the article degrades.

  • Remark: Could the authors report the running time of the proposed algorithm? In this way, we can justify whether this algorithm can be applied to large-scale dataset.

Our improvement-07: The running time of algorithms are expressed as formulas in the sections where algorithms are presented.

  • Remark: The authors should address the following limitations. The first limitation is that, the related works should be grouped into two or three subsections. In the current version, the authors all merged them together.

Our improvement-08: We have divided the related work section into 4 subsections:

2.1. Situation awareness

2.2. The Disaster Situation Awareness Problem

2.3. UAV’s and Data Fusion Techniques

2.4. Advanced Air Mobility

  • Remark: The proposed algorithm still can be improved if the ideas in the following papers are explored, i.e., "Simultaneous localization and mapping (slam) and data fusion in unmanned aerial vehicles: Recent advances and challenges", "Semantic frameworks to enhance situation awareness for defence and security applications", "Deep learning-based face detection and recognition on drones", and "Aerial orthoimage generation for UAV remote sensing‏". The authors are encouraged to discuss them in the revision.

Our improvement 09: We cite these articles in Section 2 and elaborate on them within the context of our article.

  • Remark: The author should show us a table including detail information + about source, scale of each data used.

Our improvement-10: In Appendix, we present 4 tables illustrating the characteristics of the data sources considered.  See also Remark-03.

  • Remark: The "Introduction" section should be more concise.

Our improvement-11: We have revised the Introduction section as requested.

  • Remark: The Methodology section has far too few details.

Our improvement-12: To this aim, a separate section 3.2. is introduced.

  • Remark: Please add the discussion part.

Our improvement-13: A separate Section 9 Discussion, elaborates on the article.

  1. Remark: Please reinforce the understanding of the correlation between your digital disaster response system and the AAM domain. It is obviously existing, just please improve and make more obvious.

Our improvement-14: We have emphasized this in several sections.

Section 2.4 Advanced Air Mobility and Enhancing Disaster Situation Awareness,

Section 8 Generalization of the Analysis and Synthesis Approach to UAS-based Data Fusion, and

Section 10 Future Work,

Explicitly discuss this correlation.

Reviewer 2 Report

After reading it and doing a literature review, I recommend accepting it in its current version since this reviewer finds extensive novel work. To support my recommendation, please find a detailed description of the following questions:

 1. What is the main question addressed by the research?

The authors worked on a very important topic, such as disaster situation awareness, and proposed a model-based analysis and synthesis framework to assess and recommend data fusion alternatives. With this manuscript, the authors contribute to the knowledge by introducing a model-based framework for analyzing, selecting, and optimizing data sources dedicated to earthquake disaster management. Apart from this, an object-oriented model is defined to represent geographical areas, and consequently, the effect of alternatives of data fusion can be assessed. Finally, algorithms are defined to synthesize the optimal fusion of data sources.

2. Do you consider the topic original or relevant in the field? Does it address a specific gap in the field?

The reviewer finds that the manuscript is an original study in the field of earthquake disaster management since, to the best knowledge of this reviewer, there is a lack of studies addressing all the issues detailed in this manuscript, such as the model-based framework. Consequently, the contribution of this manuscript addresses an important gap in the field of disaster management.

3. What does it add to the subject area compared with other published material?

To the reviewer's best knowledge, the manuscript adds a novel framework that can improve the earthquake disaster management field. Examples of applications are detailed and discussed in the manuscript.

4. Are the conclusions consistent with the evidence and arguments presented, and do they address the main question posed?

The reviewer considers that the conclusions are consistent with the arguments presented in the body of the manuscript. The manuscript is well-developed, and no language problems have been detected by this reviewer.

5. Are the references appropriate?

The references are appropriate.

Author Response

Thank you for the review. 

Reviewer 3 Report

It is evident that the manuscript does not meet the criteria for novelty and completeness expected of papers in Drones. My recommendation is therefore to reject this manuscript.

1.      Please refine your manuscript carefully by considering grammatical issues, avoid redundancy and bias.

2.      The author should show us a table including detail information about source, scale of each data used

3.      The "Introduction" section should be more concise. Could you summarize the references in a logical way

4.      The Methodology section has far too few details.

5.      please add the discussion part. It is important to consider some previous publications and enhance your results with these previous reports.

Please refine your manuscript carefully by considering grammatical issues, avoid redundancy and bias.

Author Response

[Drones] Manuscript ID: drones-2458493

Thank you very much for giving a detailed review of our submitted paper for the Drones special issue. We have considered the remarks seriously and worked on them carefully. We have improved our paper accordingly and realized the following improvements:

Our improvement-01:

We have emphasized contributions/novelties in the last revision as follows:

Ä°n introduction, the following paragraph is written to emphasize the contributions/novelties:

The contributions/novelties of this article are as follows. … (Please refer to the article).

In Section 11, contributions/novelties are emphasized by writing the following paragraph:
We will now elaborate on this article from the perspective of the contributions/novelties by referring to the research questions formulated in Section 3. …. (Please refer to the article).

Our improvement-02:

We have thoroughly checked the article for typos/grammar errors according to the US English spelling rules. Thank you for emphasizing this issue.

Our improvement-03:

Our data references are the characteristics of the data sources, which are presented in Appendix. In addition, we have added the following item in Section 9:

Data sets and incorrect assumptions of the effectiveness, cost, and timing values of data sources: … (Please refer to the article).

Our improvement-04: We have revised the Introduction section as requested.

Our improvement-05: To this aim, a separate section 3.2. is introduced.

Round 2

Reviewer 1 Report

All my comments have been thoroughly addressed. It is acceptable in the present form.

Author Response

Thank you for your review. We appreciate.